# Tree crickets optimize the acoustics of baffles to exaggerate their mate-attraction signal

Natasha Mhatre[1†‡*], Robert Malkin[1†§], Rittik Deb[2†], Rohini Balakrishnan[2], Daniel Robert[1]

[1]School of Biological Sciences, University of Bristol, Bristol, United Kingdom; [2]Centre for Ecological Sciences, Indian Institute of Science, Bangalore, India

**\*For correspondence:**
natasha.mhatre@gmail.com

[†]These authors contributed equally to this work

**Present address:** [‡]Department of Biological Sciences, University of Toronto at Scarborough, Scarborough, Canada; [§]Faculty of Engineering, University of Bristol, Bristol, United Kingdom

**Competing interests:** The authors declare that no competing interests exist.

**Abstract** Object manufacture in insects is typically inherited, and believed to be highly stereotyped. Optimization, the ability to select the functionally best material and modify it appropriately for a specific function, implies flexibility and is usually thought to be incompatible with inherited behaviour. Here, we show that tree-crickets optimize acoustic baffles, objects that are used to increase the effective loudness of mate-attraction calls. We quantified the acoustic efficiency of all baffles within the naturally feasible design space using finite-element modelling and found that design affects efficiency significantly. We tested the baffle-making behaviour of tree crickets in a series of experimental contexts. We found that given the opportunity, tree crickets optimised baffle acoustics; they selected the best sized object and modified it appropriately to make a near optimal baffle. Surprisingly, optimization could be achieved in a single attempt, and is likely to be achieved through an inherited yet highly accurate behavioural heuristic.
DOI: https://doi.org/10.7554/eLife.32763.001

## Introduction

Animals produce conspicuous intraspecific communication signals whose intensity is maximised through sexual selection. Several insect species, including tree crickets, use sound to advertise for mates. The louder the call, the farther it travels and the more detectable and attractive it is to listening females (*Farris et al., 1997*; *Forrest, 1991*; *Mhatre and Balakrishnan, 2007*). Several strategies exist to enhance the radiation of acoustic energy (*Bennet-Clark, 1999*), but the principal strategy insects use is to exploit the mechanical resonance either of adapted body parts, such as wings, or tymbal organs (*Mhatre et al., 2012*; *Sueur et al., 2006*), or of their immediate environment as mole crickets do with their calling burrow (*Daws et al., 1996*).

Tree crickets use a unique and distinct strategy to enhance the reach of their call; they make and use their own acoustic baffles (*Forrest, 1991*; *Prozesky-Schulze et al., 1975*; *Forrest, 1982*). Tree crickets, like other crickets, produce sound by rubbing raised forewings together and setting them into resonant vibration (*Mhatre et al., 2012*). The sound that radiates from either side of the vibrating wing, however, is of opposite phase; positive pressure or compression in the direction of the wing's motion and rarefaction or negative pressure on the other side (see *Video 1*). The wing's sound production capability is thus made inefficient by 'acoustic short-circuiting', whereby the two opposing sound waves emanating from the wing destructively interfere at its edges, thus reducing the total radiated sound pressure (*Forrest, 1991*; *Prozesky-Schulze et al., 1975*; *Forrest, 1982*) (*Video 1*). This short-circuiting is particularly potent in tree crickets because their wings are small (~9 mm in length) compared to the wavelength of their call (~110 mm at 3.1 kHz).

One way to circumvent the limitation is to separate the front and back faces of the sound source (the resonating wings) thus precluding deleterious interaction between the two pressure waves. A

**eLife digest** Male tree crickets produce sounds at a specific pitch to attract females. The louder the call, the further the sound travels and the more females he can attract. But making loud sounds is difficult for small animals like insects.

To produce sounds, tree crickets rub their wings together and set them into vibration. As the wings vibrate, their motion creates changes in the surrounding air pressure, which is perceived as sound. As the wings move forwards, they compress the air in front of them and thin the air behind them, working much like the membrane of a loudspeaker. However, when the compressed and thinned air meet at the edges of the wings, the sound cancels out. This problem is known as acoustic short-circuiting, and the smaller the wings, the larger this effect and the less efficient the broadcast of sound becomes.

Tree crickets overcome acoustic short-circuiting by making baffles, for which they cut a hole near the centre of a leaf. The cricket then sings from inside this hole with its wings flat against the leaf surface, so that the sound has to travel to the leaf edge before short-circuiting. Not all baffles work equally well though, and scientists are interested to know whether tree crickets know how to make the best possible baffle to attract more females.

To find out what makes an ideal baffle, Mhatre et al. first measured the wing vibrations and sounds of real tree crickets, and used them to simulate a cricket singing from different baffles. From these tests, three simple rules emerged that led to the best baffle: use the largest available leaf, make a hole the size of the wings, and place it at the centre of the leaf.

Mhatre et al. then discovered that the crickets did not make a baffle every time – only when the leaves were large enough. This suggests that rather than being solely 'robotic' in their behaviour and the use of objects, insects can behave flexibly. When faced with a choice between two leaves, the crickets followed the same three decision rules that the scientists had discovered, and achieved near optimal baffles.

Insects are thought to only be able to gradually improve an object or behaviour, but rarely to optimize it. However, the discovery that tree crickets can make optimal acoustic baffles in a single attempt means that we are only beginning to unravel the underappreciated abilities of insects. An enticing next step will be to see whether the creation of baffles could be considered as tool-making.

DOI: https://doi.org/10.7554/eLife.32763.002

baffle provides exactly such acoustic isolation (*Video 1*). In effect, male tree crickets cut a hole in a leaf to make a baffle. They then place their head and forelegs through the hole and call with their wings parallel to the leaf surface and centred within the hole (*Figure 1A*, *Video 2*). The leaf surface area provides acoustic isolation between the two sides, increasing call loudness (*Forrest, 1991*; *Prozesky-Schulze et al., 1975*). Given that the only function of the baffle is to prevent acoustic short circuiting, and not to resonate with the call, a tree cricket could use any leaf, or flat surface for this purpose.

Here, we show that not all leaves confer equal benefits to sound radiation. Baffle design has a profound effect on radiation efficiency, and tree crickets can select the best leaf to make a near optimal baffle. We show that we can quantitatively predict the relationship between calling effort (measured as wing vibration velocity), baffle design, and the resulting sound using biomechanical and acoustical measurements, and finite element analysis (FEA). Using FEA models,

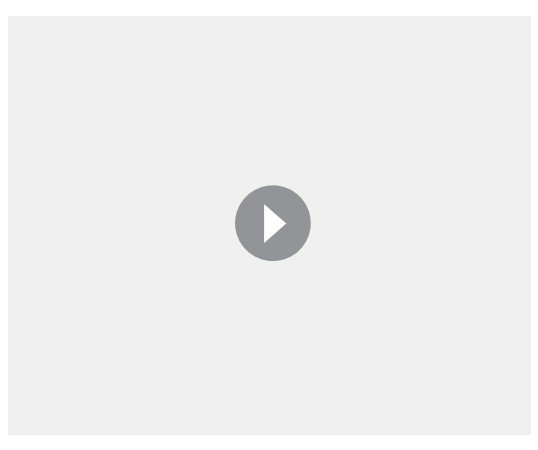

**Video 1.** Finite element analysis simulation of instantaneous sound pressure radiated by unbaffled and baffled calling cricket.
DOI: https://doi.org/10.7554/eLife.32763.003

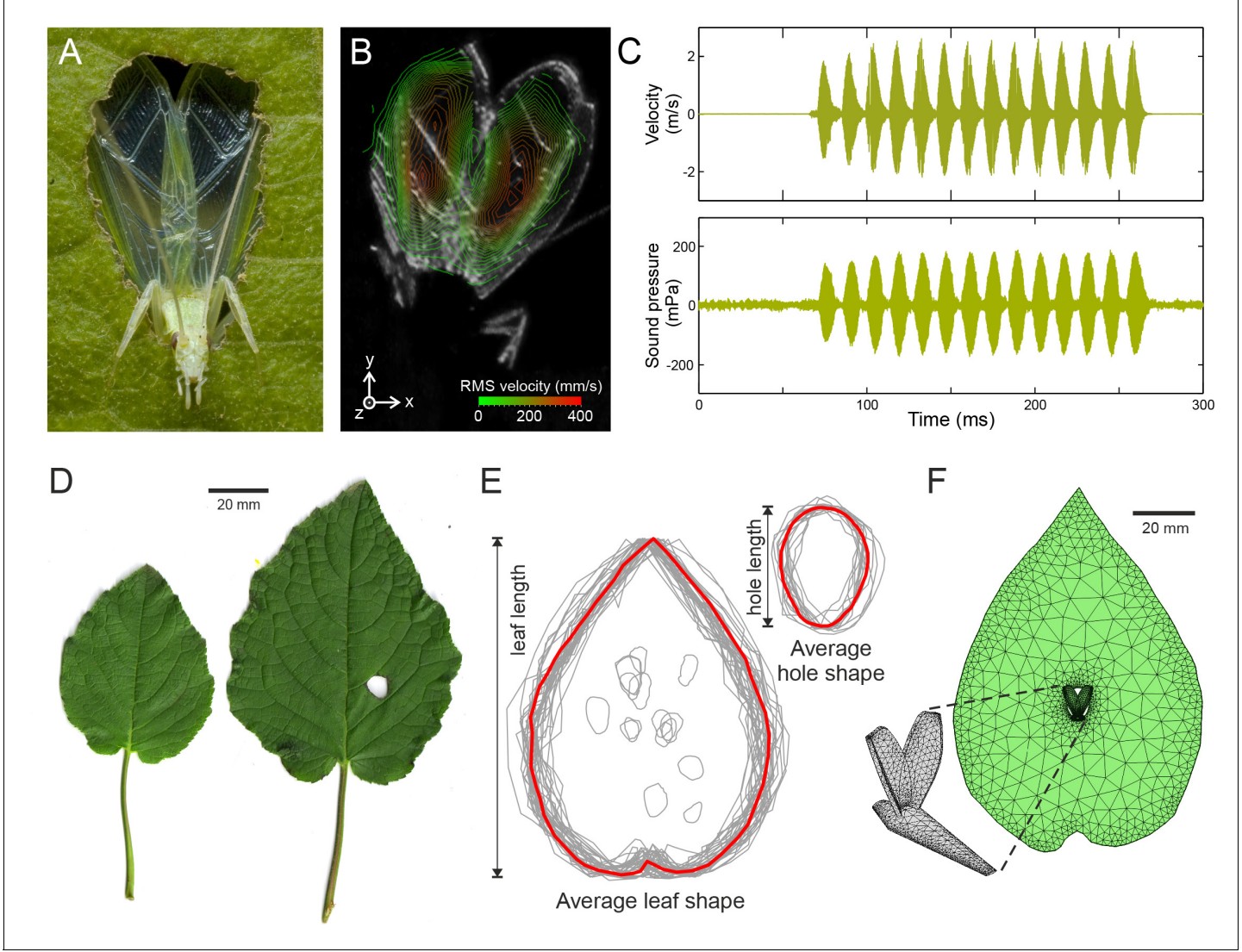

**Figure 1.** Tree cricket calling and baffle-use behaviour. (A) A wild male *Oecanthus henryi* calling from a baffle (photograph reproduced from [*Mhatre et al., 2012*]). (B) RMS wing velocity (in the z-direction) of a male calling from a baffle hole (see *Figure 1—figure supplement 1* for details). (C) Time-resolved wing velocity measured at the location of highest velocity (upper trace), and sound pressure level 200 mm from the cricket (lower trace). Peak velocity averaged 1.8 ± 0.6 m·s$^{-1}$, peak displacement reached 95.67 ± 29.54 μm, and peak accelerations of 4629 ± 1733 g were measured. This gave rise to peak SPLs of 108.8 ± 48.5 mPa (74.0 ± 3.9 dB). (D) Examples of host plant leaves (*Hyptis suaveolens)* offered to a male in choice experiments. The larger leaf shows a cricket-made baffle hole. (E) Outlines of all leaves and baffle holes in the choice experiment normalized for size and overlaid to calculate the average leaf and baffle hole shapes (red outline). (F) Finite element model of an average shaped baffle with detail showing cricket model. Model baffle hole is shown at the leaf centre, but can be arbitrarily positioned across the leaf surface.

DOI: https://doi.org/10.7554/eLife.32763.004

The following figure supplements are available for figure 1:

**Figure supplement 1.** Experimental set-up for vibrometry of calling tree crickets.
DOI: https://doi.org/10.7554/eLife.32763.005

**Figure supplement 2.** Finite element model of cricket calling from idealized and realistic baffles.
DOI: https://doi.org/10.7554/eLife.32763.006

we calculate and test the efficiency of tree crickets calling from different natural baffle designs, accounting for the complex geometries of cricket wings, their self-made baffles, and the leaves crickets chose to call from.

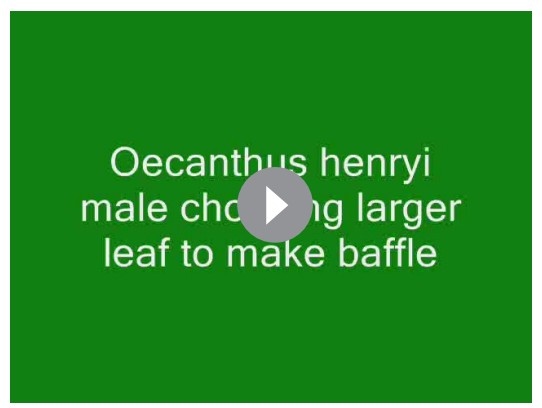

**Video 2.** Video of male *O. henryi* making a baffle and calling from it.
DOI: https://doi.org/10.7554/eLife.32763.007

## Results

### Calling effort and wing speed

To characterize calling effort, wing vibrations during calling were measured using scanning laser Doppler vibrometry. Call was recorded simultaneously at a distance of 200 mm (*Figure 1A*, *Figure 1—figure supplement 1*). Measurements were made from freely behaving animals who sang when offered pre-made paper baffles (*Figure 1B,C*, Materials and methods). The sound pressure produced by a vibrating structure is dependent on its space and time averaged velocity. It also depends on the size of the radiator and the wavelength of the sound being produced and is captured by a constant called the radiator resistance (*Hambric and Fahnline, 2007*). In tree crickets, the average velocity over the entire wing surface for a fixed period of one second was 0.13 ± 0.06 m·s$^{-1}$, and it produced an average sound pressure level (SPL) of 22.8 ± 11.0 mPa (60.4 ± 4.1 dB). Wing velocity was thus related to sound pressure level by a constant of 0.18 ± 0.03 Pa/m/s (all measures are given as mean ±SD, n = 5).

### Baffle design and efficiency

To better understand the acoustic benefit offered by baffles, FEA models were developed of a cricket calling from a baffle identical to that used in the vibrometry experiment (*Figure 1—figure supplement 2*). Vibration measurements served as inputs for FEA models and were used to compute the radiated SPL. Models predicted a bi-lobed acoustic field, consistent with previous acoustic measurements (*Figure 2*) (*Forrest, 1991*; *Prozesky-Schulze et al., 1975*; *Forrest, 1982*). A male calling in free space with average wing velocity was predicted to produce an SPL of 14.6 mPa (57.2 dB at 200 mm) (*Figure 2A*), commensurate with field measurements (*Deb and Balakrishnan, 2014*). The virtual oval baffle increased SPL at the same distance to 24.9 mPa (61.9 dB) (*Figure 2B*), consistent with the actual sound pressures recorded in the experiments (22.8 ± 11.0 mPa or 60.4 ± 4.1 dB, n = 5).

Real baffles, however, vary given that crickets encounter leaves of different sizes and produce dissimilar holes. Hence, baffles are also expected to vary in their capacity to enhance call amplitude. Larger leaves and cricket-size baffle holes are thought to make the best baffles(*Prozesky-Schulze et al., 1975*). Using FEA, we produced a quantitative comparison between different baffle designs using a metric we defined, sound radiation efficiency (SRE). Males are expected to maximize the reach of their mate attraction calls per unit calling effort. Therefore, we defined SRE as the average absolute sound pressure around the calling cricket over a fixed volume normalised to wing vibration amplitude (expressed in mPa/m•s$^{-1}$, Materials and methods). SRE was calculated only for the range of leaf and hole sizes that encompasses natural variation (*Figure 3A*, *Figure 3—figure supplement 1*). This could also be considered an index of male conspicuousness in terms of active acoustic space(*Deb and Balakrishnan, 2014*; *Mhatre and Balakrishnan, 2006*), and all else being equal, the volume of each male's active space is covariant with SRE.

We found that SRE increased with leaf size but became asymptotic for leaves longer than 100 mm, i.e. longer than most natural leaves (*Figure 3A,B*). Thus, males choosing larger leaves would always make better baffles under natural conditions (*Figure 3A*). FEA also predicted that a baffle hole of length ~10 mm, the same as the model wing length (and the size of a typical male cricket wing), was optimal (*Figure 3A*). The maximum SRE of 386.3 mPa/m•s$^{-1}$ was achieved by a baffle made with a leaf 110.2 mm long, with a centrally located hole of length 9.7 mm.

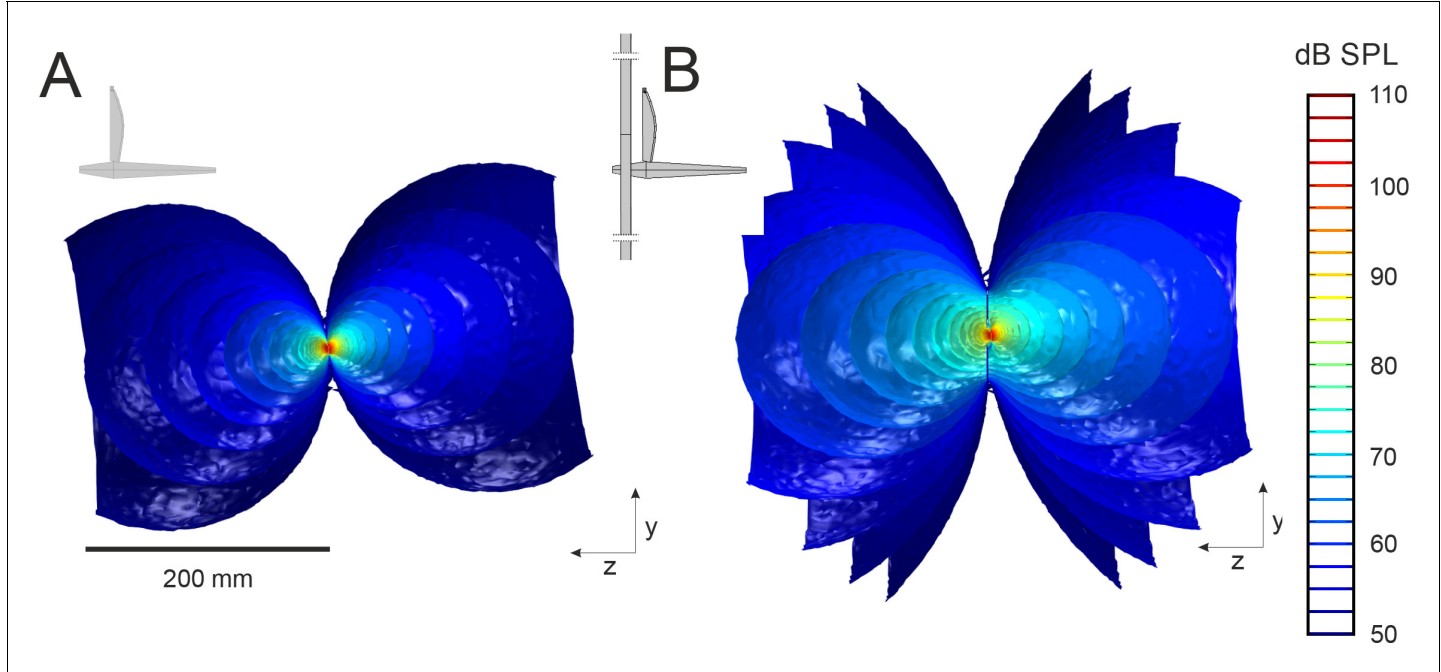

**Figure 2.** The sound field predicted by finite element modeling from an unbaffled and baffled cricket. A y-z plane cut through the 3D sound field radiated by the virtual FEA cricket (**A**) calling in free-space and (**B**) through an oval baffle (baffle length x width: 69 × 41 mm; hole length x width: 17 × 11 mm). The cricket's body aligns with the z-axis and the wing-plane is in the x-y plane as depicted in the inset. The pressure field is represented by iso-amplitude surfaces that join all points in space with identical dB SPL. Baffling reduces acoustic short-circuiting and enhances SPL around the hole, resulting in higher far-field radiation.

DOI: https://doi.org/10.7554/eLife.32763.008

The following figure supplement is available for figure 2:

**Figure supplement 1.** Effect of baffle hole position on the shape of the resulting sound field.

DOI: https://doi.org/10.7554/eLife.32763.009

## Optimization of baffle acoustics

Having constructed a geometrically-resolved map of SRE for feasible baffles, we investigated whether male tree crickets optimized the acoustics of their baffles. SRE with a baffle of any size is higher than unbaffled calling (*Figure 3A,B*), and the expectation is that males should always baffle given an opportunity. In natural populations observed over three field seasons, however, the probability of finding a baffling male was as low as 26% (27 of 105 males observed calling) and while baffling males tended to be on larger leaves, there was wide variation in leaf size (*Figure 3—figure supplement 1A*). Thus baffled-calling is not a stereotyped and obligate calling behaviour; tree cricket males must decide whether to call with or without a baffle.

In the wild, large leaves that can make acoustically optimal baffles are extremely rare and search costs are likely to be high (*Figure 3B*). However, if we eliminate search costs, the propensity to make a baffle may increase with leaf size, a hypothesis that we tested under controlled laboratory conditions. Males were offered leaves of four size classes, one leaf at a time, over four nights. Each night, they could choose either to make a baffle or to call from the edge of the leaf. In this situation, in the absence of search costs, calling males were only balancing the effort expended in making a baffle against the increased SRE. Baffling probability varied depending on the leaf size (Cochran's Q: 63.9, df = 3, p<0.001, *Figure 3B*) and was significantly different between extra-large and large leaf sizes (Mc-Nemars $\chi^2$ = 4.9, df = 1, p=0.02), large and medium leaf sizes (Mc-Nemars $\chi^2$ = 15.05, df = 1, p<0.001) and medium and small leaf sizes (Mc-Nemars $\chi^2$ = 5.14, df = 1, p=0.02). The larger the leaves, the more likely the crickets were to make a baffle (*Figure 3B*). However, the probability of making a baffle approached 50% only with leaves between 61 and 69 mm long, where SRE increased by 54%. Even with the largest leaves, baffling probability did not exceed 67%, and 33% of

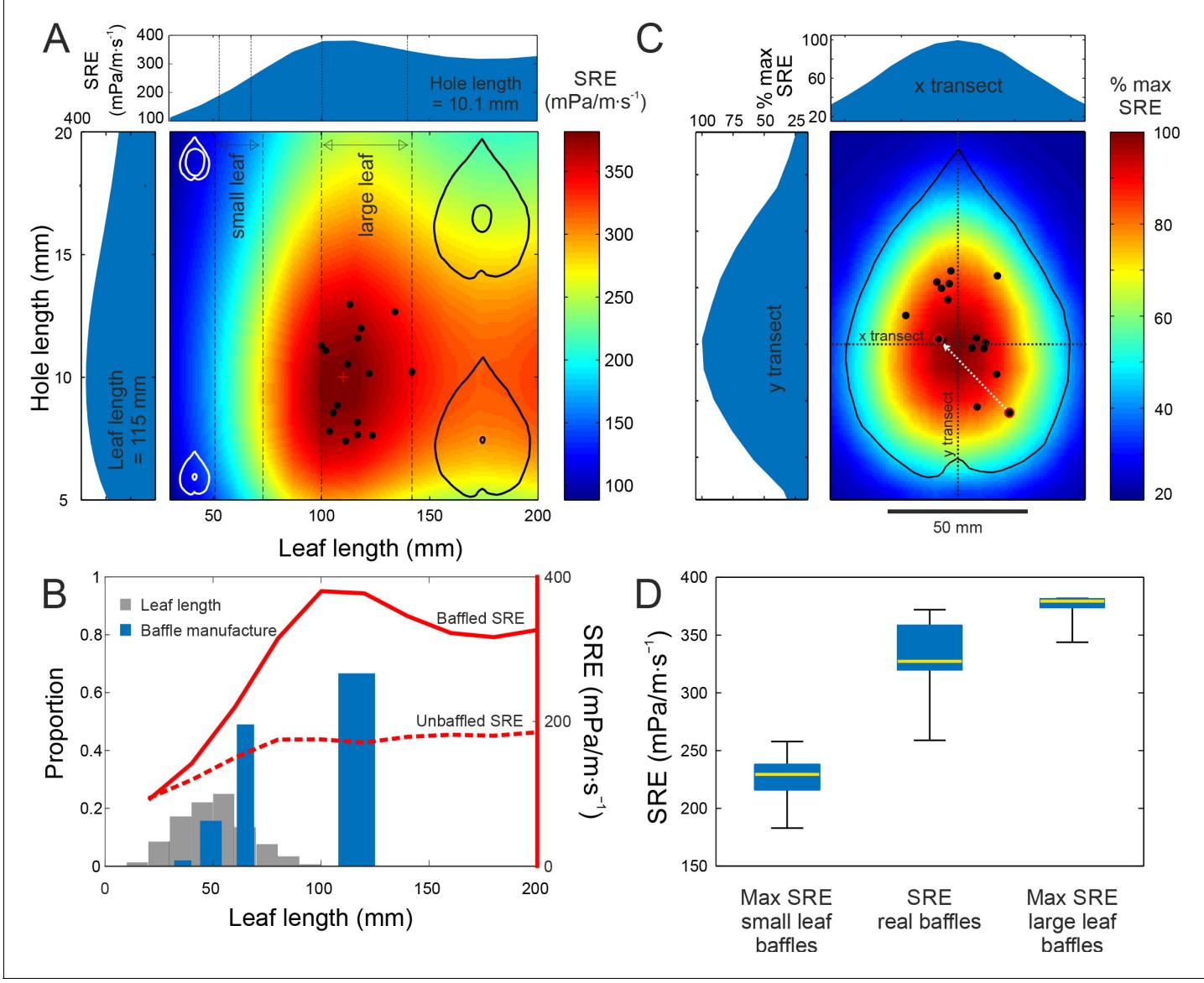

**Figure 3.** Baffle design efficiency and optimization behaviour. (**A**) FEA predicted SRE representing the effect of different leaf and hole size combinations (cartoons not to scale), overlaid with data from cricket-made baffles (black dots). Leaf and hole lengths are used as a proxy for size. In nature, as in our models, leaves retain their aspect ratio at different sizes, and length is an adequate measure. Vertical stippled lines mark the ranges of small and large leaf sizes offered to the crickets during choice experiments. Side graphs show SRE at average leaf size (115 mm) and average hole size, which is also the optimal size (10.1 mm). The red cross marks the optimal baffle. (**B**) Blue bars depict the proportion of males that made baffles on different leaf sizes in a no-choice experiment. The width of the bars depict the size range of the leaves in that size class. Grey bars depict the distribution of natural leaf sizes by depicting the proportion of leaves that fall into different size classes (N = 570 leaves). The two red lines depict the SRE associated with baffled calling (solid line), and with unbaffled calling from the leaf edge (wings parallel to the leaf surface, stippled line) at different leaf sizes. (**C**) Baffles created on real leaves were normalized to the average large leaf (length = 115 mm) and overlaid onto a map showing the percent SRE at different hole positions in relation to the maximum. Percent maximum SRE along horizontal and vertical transects are shown on side graphs. One male made two holes, first an eccentric, suboptimal one, followed by another at a more efficient location (white arrow). (**D**) Distributions of maximum SRE achievable on small and large leaves offered to the males, with SRE distributions of real baffles made by males (n = 15). Box and whisker plot depicts median (red line), the 25th and 75th percentile (blue box) and the range (whiskers).

DOI: https://doi.org/10.7554/eLife.32763.010

The following figure supplement is available for figure 3:

**Figure supplement 1.** Baffle making behaviour and its natural context (A) Box-plot showing the distributions of leaf size used by wild non-bafflers and by baffling males in the field.

DOI: https://doi.org/10.7554/eLife.32763.011

the males missed the opportunity to more than double their SRE and thus their active acoustic space (*Figure 3B*).

Next, we tested whether males preferred larger leaves or whether they could also select the best leaf when given a choice. Tree crickets could indeed select the leaves that yielded higher SREs. 19 males were presented with a choice between a small (61.2 ± 4.8 mm) and a large leaf (115.2 ± 11.7 mm) of their host plant (*Hyptis suaveolens*). Males always selected the leaves yielding the higher SRE. All males that made a baffle (15 of 19) used the larger leaf. The holes they made were also acoustically optimal in size (10.0 ± 1.9 mm, n = 15). Thus, given the opportunity males would indeed optimize the SRE of their baffles in terms of both selecting the appropriate materials, i.e. the largest leaf and then modifying it appropriately i.e. cutting a hole of acoustically optimal size (*Figure 3A*).

Males, however, did not make holes at the exact centre of leaves (*Figure 1E*). We used our FEA to further explore whether eccentric hole positions came at a cost to SRE. An SRE map was constructed by varying hole position across an average leaf in the large size class. This map revealed that SRE was highest at the leaf centre and decreased for eccentric hole positions (*Figure 3C*, *Figure 2—figure supplement 1*). Thus, eccentric hole positions come at a cost to SRE. A central-hole baffle on the smaller leaf would have achieved an SRE of 226.3 ± 20.9 mPa/m•s$^{-1}$. On the large leaf, it would achieve an SRE of 375.1 ± 11.1 mPa/m•s$^{-1}$. The radiation efficiency of the real baffles made by the crickets was 332.2 ± 30.3 mPa/m•s$^{-1}$ i.e. 88.6 ± 7.6% of the maximum achievable SRE with a skew towards higher efficiencies (*Figure 3D*) (mean ± SD, range = 72.8% to 97.7%, n = 15 all measures).

Although theoretically optimal, a central hole position is impractical as it cuts through the midrib. This compromises the leaf's structural rigidity and its main water transport system, causing it to wilt, thus reducing the lifetime of the baffle. Additionally, cutting through the rigid midrib may be mechanically difficult. Indeed, closer observation of baffle holes (*Figure 3—figure supplement 1B*) showed that males never cut through major leaf veins.

## Progressive versus single shot optimization

Surprisingly, most males made only one hole and thus made a baffle in a single attempt, without any progressive trial and error optimization of baffle design. There was, however, one exception, one male made two holes (*Figure 4A*). The first hole was far from the centre and achieved 72.1% of the maximum possible SRE. The male then made another hole closer to the centre which on its own would have achieved 96.9% of the maximum SRE.

Why did the other males make no effort to boost their call amplitude further? For instance, mole crickets gradually restructure their burrows to bring burrow resonance closer to call frequency (*Daws et al., 1996*; *Bennet-Clark, 1987*). Baffles however, are unique and cannot be 'edited' unlike cavities; once a hole has been cut in a leaf, it cannot be erased or repositioned. Only a new hole can be made. Here we show that cutting a new hole entails the risk of making a sub-optimal baffle by reducing current SRE through increased acoustic short-circuiting. We studied the effect of a second hole on SRE by simulating a scenario where an eccentric hole was abandoned and a second, optimally centred hole was constructed. The map depicts at each primary hole position, the percent change in SRE if the male abandoned that position and made a centrally positioned secondary hole (*Figure 4B*).

A second baffle hole would accrue a large increase in SRE only if the primary hole was near a leaf margin and produced very low SRE to begin with (*Figure 4B*). The only two-holed baffle observed was from a male that made an unusually eccentric primary baffle hole (highlighted in *Figure 4B*). In this example, making a more central hole improved SRE by 37.5%, despite increased acoustic short-circuiting. Interestingly, our simulations predict that the more central the primary hole, the less beneficial a secondary baffle hole becomes and in the central zone of the leaf, a second hole becomes deleterious to SRE (*Figure 4B*). This is because the second hole would fuse with the primary hole effectively enlarging it to a suboptimal size (*Figure 3A*). Our observations show that the first holes made by males are already close to the centre; if they made a second central hole, six males would have decreased SRE by 6.73 ± 2.77% and the other nine males stood to increase SRE but only by 19.17 ± 11.15% (*Figure 4B*). This finding supports the notion that crickets position their baffle holes with geometric optimality, but also avoid jeopardising this optimality. This may be why large, and sought-after, leaves only ever have one baffle hole. Near-optimal SREs are achieved by most males

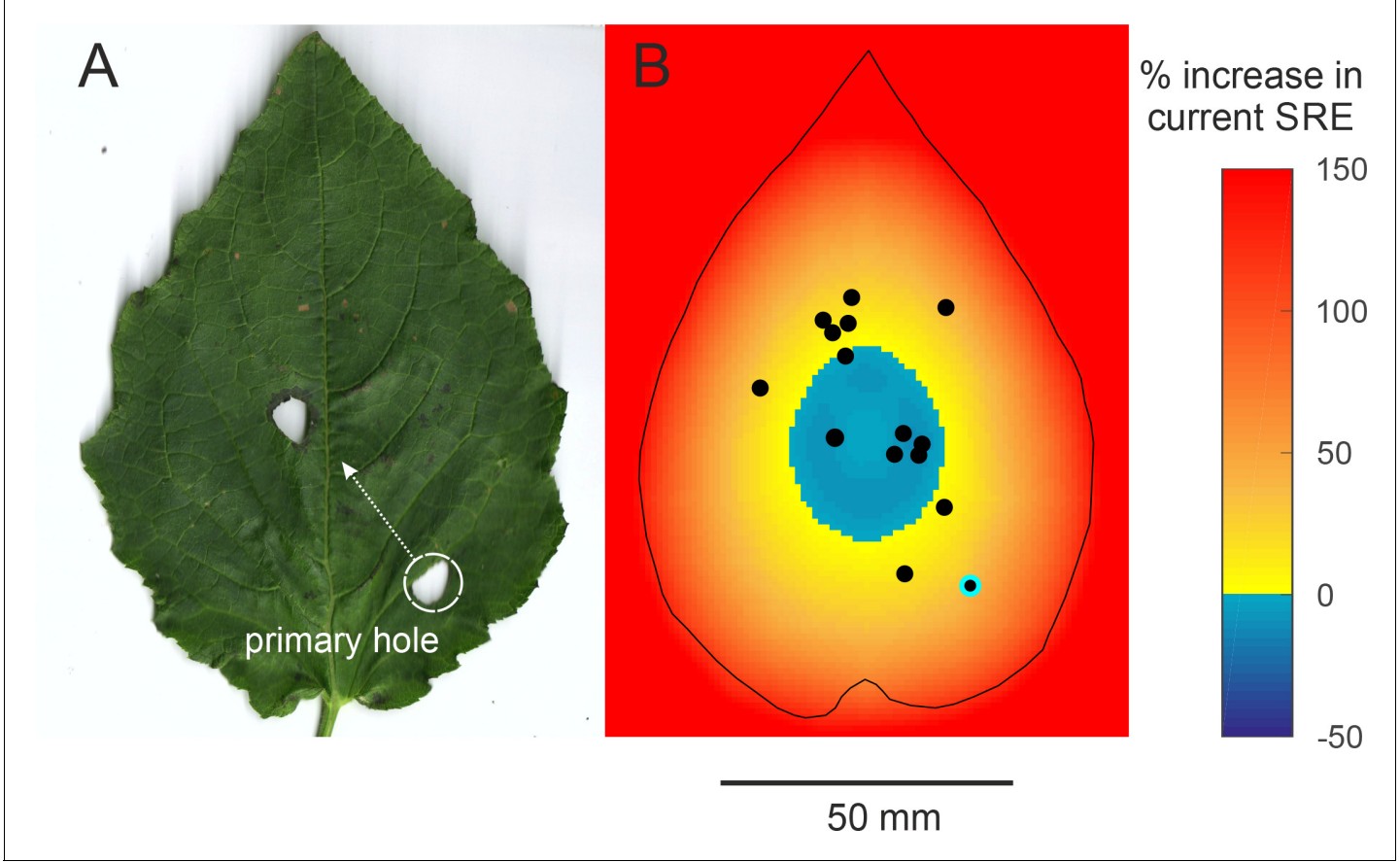

**Figure 4.** Efficiency of two-holed baffles (A) The baffle of a male who made a very eccentric primary hole and then cut another hole at a more central location. The SRE of a two-holed baffle such as this is difficult to predict *a priori*. A central baffle hole is more efficient, but two holes allow more acoustic short-circuiting. (B) An SRE map of a two-holed baffle was made to address this question. In this scenario, the primary hole can be positioned anywhere on the leaf but the second hole is always at the optimal central position. The colour on the map depicts the percent increase in SRE that a male at that position can accrue by making a second optimally positioned hole. SRE increases only if the primary hole was eccentric; if the primary hole was close to the optimal position, however, making a second hole in effect reduces the baffle's SRE. The dots depict the positions of the holes of real males. The primary hole of the only male with the two-holed baffle is highlighted.

DOI: https://doi.org/10.7554/eLife.32763.012

at the first baffling attempt (*Figure 3D*) and most attempts at improvement can be predicted as being futile (*Figure 4B*).

## Discussion

### The mechanism for baffle optimization

The question naturally arises as to how tree crickets make acoustically optimal baffles in a single attempt? In the tree cricket genus *Oecanthus*, baffle making is common (*Forrest, 1991*; *Prozesky-Schulze et al., 1975*; *Forrest, 1982*) and expressed by many individuals within the species, suggesting that it is an inherited trait. Thus, the optimization procedure may also be 'hard-wired', rather than learned and may have developed *via* sexual selection.

We surmise that tree crickets accomplish acoustic optimization using hard-wired heuristics or 'rules of thumb' (*Hutchinson and Gigerenzer, 2005*; *Cross and Jackson, 2005*), which require them to make a series of choices. If a male decides to make a baffle, three simple rules can produce an optimal baffle in a single attempt: (i) find the largest available leaf, (ii) place the hole as close to the centre of the leaf as possible and (iii) cut a hole that can just accommodate wings (*Figure 3*, *Figure 3—figure supplement 1B*). These rules encode sufficient information to capture the shape of

SRE landscape within baffle design space into a simple, yet high-accuracy heuristic that always produces the baffle with the highest possible SRE.

The accuracy of this heuristic is higher than observed in the progressive optimization procedure observed in mole crickets burrows which always perform just sub-optimally (*Daws et al., 1996*). At 3 kHz, the burrow resonance is always higher than call frequency at 2.5 kHz and the actual burrow dimensions do not fit optimal dimensions for sound radiation at either frequency (*Daws et al., 1996*). The explanation for this sub-optimal performance lies in Weber's law, whereby sensory systems cannot distinguish between small differences in signal amplitude or frequency at high signal amplitudes due to receptor saturation and habituation (*Imaizumi and Pollack, 2001*; *Imaizumi and Pollack, 1999*; *Givois and Pollack, 2000*). This neurophysiological limitation is likely to be a common driver of suboptimal performance in any method that uses feedback for optimization (*Nachev et al., 2017*).

It is possible that the tree crickets learn the SRE landscape, however, we believe that this is unlikely since the opportunity for learning is small due to life-history and time constraints. Tree crickets call only as adults, and for only a few hours each day. Finding and testing each leaf size with every baffle position to find the optimal solution, especially with large leaf sizes being so rare, would require a formidably large number of learning trials and considerable memory. Another possibility is that they can abstract these general rules from a few trials. While we believe that inherited heuristic optimization is a more parsimonious explanation, however, repeating these experiments with naïve males would provide a more definitive answer.

## Heuristic optimization in object manufacture

Heuristics are, in essence, rational search methods (*McFarland, 1991*). When an object or behaviour has different efficiencies depending on a set of parameters, heuristics guide the search for the optimal position in the landscape described by these efficiencies (*McFarland, 1991*). Therefore, using a heuristic demonstrates the implicit or explicit knowledge that a behaviour can be accomplished in multiple ways and that each way differs in performance. Therefore heuristics, whether invertebrate, or vertebrate including humans, inherited or learned, imply not only an expectation of performance variability, but also choice and behavioural flexibility. One context in which this may be important is tool use; the conventional view is that invertebrates generally inherit tool use behaviour (*Hunt et al., 2013*) and hence have highly *stereotyped* tool use, whereas vertebrates inherit a propensity to 'innovate' tool behaviour and hence have *flexible* tool use (*Shettleworth, 2009*, *2010*). Inflexible or stereotyped tool use implies that optimization would not be possible for any invertebrate tool-users. It is suggested that baffles may be tools or borderline tools (*Bentley-Condit and Smith, 2010*; *Pierce, 1986*; *Crain et al., 2013*), therefore baffle optimization suggests that inherited mechanisms such as heuristics may enable flexibility and even optimization in invertebrate tool use.

Flexible tool use has indeed already been observed in invertebrates, first in cephalopods (*Finn et al., 2009*) and more recently, from bees (*Loukola et al., 2017*; *Alem et al., 2016*). Bees have a large capacity for learning, which enables significant behavioural flexibility, and it has now been shown that bees can be taught to use tools. They can even improve upon the tool use demonstrated to them (*Loukola et al., 2017*; *Alem et al., 2016*). Indeed, we argue that the behavioural flexibility that enables bees to improve upon demonstrated tool use emerges from a heuristic strategy. Bumble bees were trained to roll a ball into a specific position to gain a reward. The bees spontaneously used the ball nearest to the target even when they had been trained on a different ball (*Loukola et al., 2017*), an excellent example of the well-known greedy heuristic (*Cormen et al., 2009*) in which the nearest solution is the most preferred. Indeed, tree cricket behaviour shows that even inherited heuristic mechanisms may enable flexibility and even optimization, prompting a re-examination of invertebrate tool use.

## Baffle-use and sexual selection

In summary, tree cricket males that call from the edge of leaves tend to have low efficiencies (94–171 mPa/m•s$^{-1}$ depending on leaf size) and those that make baffles can achieve much higher efficiencies (91–386 mPa/m•s$^{-1}$ over same size range). Thus, by finding the largest leaf and making a near optimal baffle, a male tree cricket can increase his active acoustic area by as much as four times (12 dB), greatly increasing his chances of attracting a mate. The increase in call amplitude that

baffling confers can work in two ways. Baffling can either add to the existing variability, broadening the distribution of male call amplitudes or it can function as a compensatory mechanism: males unable to call loudly may make a baffle instead.

Behavioural experiments show that despite the advantage baffles confer, only a few animals choose to make a baffle under natural conditions. Non-host plants with larger leaves are available to the tree crickets, yet are rarely used. The rarity of baffling suggests that manufacturing a baffle may be a costly activity (*Figure 3—figure supplement 1A*). In contrast, all mole cricket males stereotypically make acoustic burrow chambers that resonate and boost call amplitude (*Hill et al., 2006*; *Hill, 1999*). The low incidence of baffling and high variability in baffle design observed in natural baffling behaviour (*Figure 3—figure supplement 1A*) are quite distinct. Baffling may be used to garner even higher mate-attraction benefits on males whose high quality or condition already allows them to make larger energetic investments. Thus, baffle making may represent a unique system in which 'good' males use an acoustically optimized object to further enhance their mate-attraction signal. Another enticing possibility is that baffle-making may have evolved as an alternative strategy allowing smaller males, or those in poorer condition, to sound bigger or louder than they actually are.

## Materials and methods

### Vibration measurement of calling crickets in the laboratory

Vibration velocities from the forewings of freely calling tree cricket (*Oecanthus henryi*) males calling from baffles were measured using a micro-scanning laser Doppler vibrometer (Polytec, PSV-400, Waldbronn, Germany) with a Polytec PSV-I-400 scanning head fitted with a close-up attachment, and digitized using the Polytec Scanning Vibrometer software (version 8.8, Polytec Gmbh, Waldbronn, Germany) through a data acquisition board (National Instruments, PCI-6110, Austin, Texas, USA).

Acoustic measurements were made simultaneously using two calibrated 1/8th inch precision pressure microphones (Brüel and Kjær, 4138, [frequency range 6 Hz to 140 kHz], Naerum, Denmark) and preamplifier (Brüel and Kjær, 2633, Naerum, Denmark). All data were collected at a sampling frequency of 128 kHz for 1.024 s. The microphones have a flat response in the measured frequency range.

All experiments were carried out on a vibration isolation table (Technical Manufacturing Corp, 784-443-12R, Peabody, Massachusetts, USA). The experimental set-up (*Figure 1E,D*) allowed the calling cricket to be oriented such that the upraised wings were normal to the path of the laser beam. The two microphones were placed coaxially ≈ 20 mm below the path of the laser beam, at a distance of ±200 mm from the center of the turn-table, hence from the position of the calling male. The laser was aligned using a video feed and it could then be remotely positioned by the vibrometer software across a grid of points placed over the wings of the calling insect. An average of 245 measurement points across the wing surface were used per animal leading to an average scan duration of just over 4 min.

Paper baffles (made using white 80 g/m$^2$ paper) were used in the study. These premade baffles were oval in shape and had a length of 68.6 ± 3.6 mm and width of 40.8 ± 2.2 mm on average (n = 5) and the holes were 17.4 ± 1.9 mm in length and 11.4 ± 1.9 mm in width (n = 5). During calling, the peak baffle surface displacements were considerably lower (1/600th) than the wings leading us to conclude that the baffles did not act as significant secondary acoustic radiators.

The sound pressure radiating from a vibrating body depends on the space-time average of the body's vibration velocity (*Hambric and Fahnline, 2007*). The peak value, which we also report in the main paper, is taken from the point of maximum deflection of the wing, as shown in *Figure 1A*. The rest of the wing does not move with this magnitude. The resulting sound is a consequence of the whole wings movement not just of this one point. If we used the peak value alone, the magnitude of wing vibration would be over-estimated. Taking an average in space resolves this issue. Since pressure is conventionally measured after integration over a fixed time period for amplitude-modulated signals, to be comparable, we also performed a time average of the vibration velocity. Hence we calculated the space-time average of the velocity in the z-direction (*Figure 1—figure supplement 1*) from all the measurement points placed over the entire surface of the wing during calling.

## Finite element analysis

Finite element analysis (FEA) models of a cricket calling in free space or on different baffles were created to investigate pressures produced by vibrating cricket wings. FEA modeling was carried out using a commercial package (COMSOL 4.3, Burlington, Massachusetts, USA). The acoustics module was used implementing harmonic analysis with the Helmholtz equation as the governing equation.

### Model geometry and vibration

The models used the following components as needed: the insect body, wings, a leaf with a hole and a large air sphere (210 mm radius) centered on the insect in which the radiated sound field was modelled (*Figure 1—figure supplement 2*)). In the model only the insect wings were in motion and hence were the only sound radiators. The body of the insect was simplified to have only the head, thorax and abdomen. A more refined representation of the insect body had negligible effect on predicted sound pressures.

Wing geometry was derived from average dimensions of tree cricket wings (n = 6). The leaf geometry was based on spatially averaged overlays of real leaves as shown in *Figure 1E*. The baffle hole in the leaf was based on similar measurements of baffle holes manufactured by real crickets (*Figure 1*). For the purposes of the model, wing vibrations were simplified and both wings were modelled as moving only in the longitudinal axis of the animal. The prescribed vibration magnitude was taken from the space-time average of the vibrometry measurements of real calling crickets. The cricket position within the baffle-hole mimicked real crickets who place their head slightly through the hole and their wings parallel to the leaf surface. The radiated instantaneous pressure distributions for unbaffled and baffled calling males are shown in *Video 2*.

In order to allow direct comparison between the calling males and the FEA models, we defined a metric, sound radiation efficiency (SRE) that we use as our figure of merit. SRE is defined as the volumetric average of the absolute sound pressure within a 210 mm radius air sphere centered on the calling cricket divided by the space-time average of the wing velocity, and has units of $Pa/m \bullet s^{-1}$.

### Boundary conditions

The wings of the cricket were given a velocity normal to their surface (coaxial with the longitudinal axis of the cricket (*Figure 1—figure supplement 2C*). The leaf and the cricket's body were given sound hard boundary conditions. The remainder of the airspace was surrounded by a perfect matched layer (PML) ensuring minimum reflections from the model domain edges. As the complex interaction between the leaf and the wing induced pressure field, and any boundary layers upon the leaf, are unknown these boundary conditions are felt to be a suitable approximation. We believe any secondary vibration of the baffle itself however to be minor, as suggested by comparatively small vibrations of the experimental baffles recorded during calling.

### Element details

3D quadratic tetrahedral elements were used to mesh the air domain and the element number was ≈ 100 k, varying slightly depending on the model. In order to ensure that the calculated pressures within the air volume were not dependent upon the fidelity of the air space discretization, a mesh convergence study was undertaken. The study indicated a minimum individual element size of ≈ 15 μm, a maximum of 1/8th of the wavelength and a PML thickness of 50 mm would ensure that the calculated pressures were independent of mesh fidelity. Further increases in mesh fidelity (of the animal, air domain and PML) would have negligible effect on the parameters of interest.

### General model parameters

There are a number of variables that describe the geometry and kinematics of the calling cricket. To explore tool optimization we first developed a 'general' model of a cricket based on observed calling behavior.

### Call frequency

The crickets in the vibrometry experiments sang at a frequency of 3.10 ± 0.19 kHz, (mean ± SD, n = 5) and this was the average value used for all models. The crickets in the choice experiment

were maintained at the appropriate temperature (26.3 ± 0.1°C, n = 15) so that they would call at a similar frequency (3.00 ± 0.07 kHz, mean ± SD, n = 9).

## Costal angle
The costal angle is defined as the angle between the dorsal field of the wing and the lateral or costal field (*Figure 1—figure supplement 2*). The angle observed in living crickets was ≈137° (n = 5) and this was the value used.

## Leaf wing distance
The leaf wing distance is the parallel in-plane separation between the leaf and the wings. The distance of real crickets could not be measured directly but was estimated from the video images during vibrometry recording to be around 1 mm and this was the value used.

## Leaf tilt angle
The angle between the leaf surface and wing surface in real crickets appeared to be 0° (both surfaces being parallel to each other) and this was the value used.

## Effect of leaf length and hole length on sound field
The combined effect of leaf and hole length was studied by modeling a matrix of combinations. Leaf length was varied from 30 mm to 200 mm at 14 mm intervals and hole length was varied from 5 mm to 20 mm at 0.8 mm intervals. The SRE for each of these matrix combinations was predicted by the FEA model, and the resulting data were linearly interpolated and presented as an image in *Figure 3*.

## Effect of baffle hole position on sound field
The effect of baffle hole position on the leaf (horizontally and vertically) on the resultant radiated acoustic field was studied on a large leaf of average length (115 mm). Baffle holes were placed at a number of positions on the leaf, creating a matrix of locations. The baffle hole was placed at 8 mm intervals in the x direction and 10 mm intervals in the y direction. SRE was predicted by the FEA models for each of the matrix positions. The same method was used when simulating a baffle with two holes. This was converted to % maximum SRE and this matrix was displayed after linear interpolation as an image in *Figure 3* and *Figure 4*. The sound field radiating from two examples, a geometrically centered hole, and a hole 30 mm from the center at an angle of 45°, is shown in *Figure 2—figure supplement 1*. The reduction in the SRE for crickets calling closer to the edge can be explained by an increase in the acoustic short-circuiting experienced close to the vibrating body where the air pressure amplitude is highest.

## Cricket calling behavior
*Oecanthus henryi* males are observed to call and manufacture baffles from *Hyptis suaveolens* leaves in the wild. To better understand their interactions with the leaves, we performed three independent studies.

## Effect of leaf size on likelihood of baffling
We first investigated the cricket's likelihood of creating a baffle in a leaf. To understand the range of leaves that may potentially be used as baffles, we measured 570 randomly selected leaves and created a leaf length frequency distribution, shown in *Figure 3B*. Data were collected from 114 plants across three field sites (50 from site 1, 30 from site 2 and 34 from site 3). Once a calling male was found on a plant, five leaves were picked at random from the plant and their maximum length and breadth was measured using a scale. For this study, the whole plant was divided into four quadrants and a leaf was picked from a randomly selected quadrant at a random distance from the centre of the plant and at a random height using previously generated random numbers.

From this, we selected 4 sizes of leaf which would be presented singly to a male tree cricket; (a) small (length: 35 ± 2 mm), (b) medium (length: 50 ± 2 mm), (c) large (length: 65 ± 2 mm) and (d) extra-large (length: 115 ± 4 mm).

51 males were presented with a leaf of randomly picked size once an evening for four consecutive evenings (random sampling without replacement). Each evening an individual male was released onto the twig of the leaf and observed for 3 hr. At the end of the experiment, it was noted if the cricket had made a baffle hole on the leaf. The results are shown in *Figure 3B*.

## Leaf size preference and creation of a baffle hole

We studied the behavior of males when a choice between a small and large leaf was available and also the size and location of a baffle hole they made. The experiment was conducted indoors in an anechoic chamber on 19 males. The setup consisted of two *H. suaveolens* leaves, small (length: 51–73 mm) and large (length: 100–142 mm) attached to a *H. suaveolens* twig pinned to a polystyrene foam base (*Video 2*). The leaves were attached at exactly the same elevation on the twig such that the petioles of the leaves were 180° apart. This creates a decision point. Both leaf stalks were covered using wet cotton to prevent the leaves from desiccating.

The experimental arena was divided into two halves (left and right) and the orientation of the big leaf (and hence the small leaf) was randomly selected on each night to correct for any directional bias shown by the animals while choosing the leaves. The set-up was covered by a transparent plastic container (150 mm height, 400 mm length and 300 mm width) so that the animals could be confined inside the arena.

For each trial, an animal was released at the base of the central *H. suaveolens* twig. Trials were conducted in complete darkness and an infra-red sensitive video camera (Sony, XR 500E, Tokyo, Japan) was used to record animal movements and choice. The trials had a cut-off time of 3 hr, within which the animal had to choose a leaf and call from it. If an animal did not call within this duration it was dropped from the experiment. Once the animal selected a leaf and consistently called from it (initially from the edge) for at least 10 min, the plastic container was removed for the rest of the night. Each animal was then recorded calling from the edge of the leaf for at least 70 pulses (200 mm from the front of the animal) and sound pressure level was measured. If the animal then made a baffle, the animal was recorded again for at least 70 pulses and sound pressure level was measured. The recordings were performed using a solid-state audio recorder (Marantz, PMD-660 [frequency range 20 Hz to 20 kHz], Kanagawa, Japan). All calls were recorded at a sampling rate of 44.1 kHz and were saved as uncompressed sound files. Sound pressure levels of the calls were measured using a calibrated sound level meter (Brüel and Kjær, 2250, Naerum, Denmark) with a ½" microphone (Brüel and Kjær, 4189 [frequency range 20 Hz to 20 kHz], Naerum, Denmark).

Experiments were conducted at a mean temperature of 26.1° C maintained using a room heater. During the experiment, the temperature of the arena was monitored with a digital thermometer (Testo, 110 Precision Thermometer, Hampshire, UK). Before experiments were started, the animals were acclimatized for 30 min to the temperature in the experimental room. Experiments were conducted in the evenings between 6:45 PM and 9:45 PM, the peak activity time for these animals. The leaf selected by the animal for making a baffle was noted. If any animal did not make a baffle within this time, it was not used for a subsequent trial, i.e. animals were tested only once. One male made two holes of nearly identical dimensions. We have, however, considered only the second hole in our analysis of hole location in order to avoid pseudo replication.

All leaves used for the trials were scanned using a flatbed scanner (Hewlett-Packard, M10005MFP, Idaho, USA), and the images were used for further analysis. After the trial, the animals were preserved in 70% alcohol and their wings were dissected under a microscope and photographed for measurement. Dimensions of baffle leaves and holes were analyzed using image processing software (ImageJ v1.43, National Institutes of Health, USA). Examples of baffle holes created are shown in *Figure 1D*). Call recordings were analyzed using commercial analyses software (Pioneer Hill Software, Spectra Plus-Professional - Version 3.0a, Washington, USA).

## Baffling and non-baffling leaf size measurements in the wild

In the field, all the baffling and non-baffling individuals in a plot were localized and the leaves from which they were calling were collected over three field seasons (two dry seasons and one post-monsoon season, 2011–2012). All the field data were collected on marked individuals. Focal individuals were not resampled in this experiment. On each night, a male calling from a baffle was found, and then all the other individuals who were calling in the same plot were found. The length of the leaves

from which these males were calling were collected. A total of 27 baffling and 78 non-baffling leaves were collected. These were then pressed separately using a standard botanical tissue collection protocol to ensure they did not change shape or size due to wilting (*Ministry of Forests Research Program, 1996*). The leaves were brought back to the laboratory and scanned using a flatbed scanner (HP Laser Jet M10005MFP, Idaho, USA) and measured using image processing software (ImageJ version 1.43). Distribution of the leaf sizes used for baffling was compared with the distribution of non-baffling leaf sizes to examine size bias in the choice of leaves used for baffling, as shown in *Figure 3—figure supplement 1* .

## Acknowledgements

The project was funded by grants from the Biotechnology and Biological Sciences Research Council UK, the UK India Education and Research Initiative and the Ministry of Environment, Forests and Climate Change, Government of India. RD was supported by a fellowship (09/079 (2199)/2008-EMR-I) from the Council of Scientific and Industrial Research, Government of India. NM was supported by a Marie Curie and a Wissenschaftskolleg zu Berlin fellowship. DR was supported by Royal Society Wolfson fellowship. We thank John McNamara for his comments and insights, and Manjunatha Reddy for help with field observation and experiments. Preliminary data from this research were presented at the Acoustics 2012 conference in Nantes, France.

## Additional information

### Funding

| Funder | Grant reference number | Author |
|---|---|---|
| Biotechnology and Biological Sciences Research Council | BB/I009671/1 | Daniel Robert |
| UK-India Education and Research Initiative | | Rohini Balakrishnan Daniel Robert |
| Ministry of Environment | | Rohini Balakrishnan |
| Council of Scientific and Industrial Research | 09/079(2199)/2008-EMR-I | Rittik Deb |
| Wissenschaftskolleg zu Berlin | | Natasha Mhatre |
| European Commission | 254455 | Natasha Mhatre |
| Royal Society | | Daniel Robert |

The funders had no role in study design, data collection and interpretation, or the decision to submit the work for publication.

### Author contributions

Natasha Mhatre, Conceptualization, Data curation, Software, Formal analysis, Validation, Investigation, Visualization, Methodology, Writing—original draft, Project administration, Writing—review and editing; Robert Malkin, Conceptualization, Resources, Data curation, Software, Formal analysis, Validation, Investigation, Visualization, Methodology, Writing—original draft, Project administration, Writing—review and editing; Rittik Deb, Conceptualization, Resources, Data curation, Software, Formal analysis, Funding acquisition, Validation, Investigation, Visualization, Methodology, Project administration, Writing—review and editing; Rohini Balakrishnan, Daniel Robert, Conceptualization, Resources, Supervision, Funding acquisition, Methodology, Project administration, Writing—review and editing

### Author ORCIDs

Natasha Mhatre (iD) http://orcid.org/0000-0002-3618-306X
Robert Malkin (iD) http://orcid.org/0000-0003-3043-6633
Rittik Deb (iD) http://orcid.org/0000-0002-8562-7034

Rohini Balakrishnan [ID] http://orcid.org/0000-0003-0935-3884
Daniel Robert [ID] http://orcid.org/0000-0002-5907-3912

### Decision letter and Author response
Decision letter https://doi.org/10.7554/eLife.32763.017
Author response https://doi.org/10.7554/eLife.32763.018

## Additional files

### Supplementary files
• Transparent reporting form
DOI: https://doi.org/10.7554/eLife.32763.013

### Major datasets
The following dataset was generated:

| Author(s) | Year | Dataset title | Dataset URL | Database, license, and accessibility information |
|---|---|---|---|---|
| Mhatre N, Malkin R, Deb R, Balakrishnan R, Robert D | 2017 | Data from: Tree crickets optimize the acoustics of baffles to exaggerate their mate-attraction signal | http://dx.doi.org/10.5061/dryad.f9011 | Available at Dryad Digital Repository under a CC0 Public Domain Dedication |

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
