## [Decision Letter]

Thank you for submitting your article "Tree crickets optimize the acoustics of baffles to exaggerate their mate-attraction signal" for consideration by *eLife*. Your article has been reviewed by two peer reviewers, and the evaluation has been overseen by a Reviewing Editor and Ian Baldwin as the Senior Editor. The following individuals involved in review of your submission have agreed to reveal their identity: Coen P.H. Elemans (Reviewer #1); Ronald Hoy (Reviewer #2).

The reviewers have discussed the reviews with one another and the Reviewing Editor has drafted this decision to help you prepare a revised submission.

The reviewers and editors concur that your manuscript is very well written and presents an excellent combination of finite element modeling, fieldwork, and carefully designed and executed lab experiments. The quality and presentation of the data and figures is high. The methods are carefully and completely described.

Summary:

This compelling paper reports how male tree crickets modify leaves by cutting a hole in the leaf at a near-optimal location to make their call better heard over longer distances by potential mates. The research integrates behavioral research with detailed acoustic simulations. The tree cricket makes its call by rubbing its wings together so that they vibrate and emit sound, the problem is that both wings are so close together that the two sound sources cancel each other out by a significant factor. By making the sound in a hole in the middle of a leaf instead, the sound of both wings gets separated by the leaf "baffle" and this much reduces the sound cancelation, hence the call carries further. Whereas we can't call cutting a hole in a leaf insect tool use, it is fascinating that an insect knows how to modify a leave to use it as an acoustic baffle and knows how to position itself in the self-cut hole to improve the audibility of its call. This remarkable finding is likely to be of broad interest to researchers studying the evolution of animal behavior, as well as behavioral ecology, neuroethology, and acoustics.

Essential revisions:

Abstract, second to last sentence and several other locations; I find the use of the word materials here misleading, because it implies the animals select from materials with different material properties. Instead it seems more precise to discuss how the animals choose between different possible baffle hole geometries.

Introduction, second paragraph and several other locations; The advertisement sounds made by tree crickets are typically not considered to be songs; Only a very lax definition of song (e.g. in Broughton 1963; "Sound of animal origin which is not both accidental and meaningless") includes these sounds. We recommend to term them calls.

Subsection “Singing effort and wing speed”, throughout source levels are reported at 200mm which is a less conventional reference distance. Please also report source levels with respect to 1 m distance if possible or discuss in the Materials and methods why 200mm is the better choice if you have a specific compelling reasoning.

Subsection “Optimization of baffle acoustics”, second paragraph. Figure 3 is somewhat confusing. The leaf distribution and probability seem to belong to the left axis and SRE to the right axis; this could be made more intuitive by a different color scheme. Also, SD error bars are missing on blue baffle manufacture bars. It might be clearer if the raw data of leaf size and probability for the individuals are added.

Subsection “Progressive versus single shot optimization”, second paragraph, mole crickets optimize their burrow structure to match call frequency resonance. This contradicts the statement or gist in the Abstract that there are no precedents of optimization in innate behavior, please discuss in the manuscript with accompanying references to frame the novity of the finding in this study better.

Subsection “Progressive versus single shot optimization”, second paragraph and subsection “The mechanism for baffle optimization”, last paragraph. Based on the evidence it cannot be fully excluded that males use the experience they gained in making baffles. E.g. just a few iterations could aid in increasing efficiency, although we would not know how they evaluate this. To be conclusive it would be helpful to control for the baffling history of the males by using naïve ones etc. This would be a rather simple future experiment presenting a very interesting finding. Please discuss your perspective on this in the manuscript for completeness and for stimulating further research. We are not asking for these controls as the contribution of the present work is sufficient to merit publication.

---

## [Author Response]

Summary:This compelling paper reports how male tree crickets modify leaves by cutting a hole in the leaf at a near-optimal location to make their call better heard over longer distances by potential mates. The research integrates behavioral research with detailed acoustic simulations. The tree cricket makes its call by rubbing its wings together so that they vibrate and emit sound, the problem is that both wings are so close together that the two sound sources cancel each other out by a significant factor. By making the sound in a hole in the middle of a leaf instead, the sound of both wings gets separated by the leaf "baffle" and this much reduces the sound cancelation, hence the call carries further. Whereas we can't call cutting a hole in a leaf insect tool use, it is fascinating that an insect knows how to modify a leave to use it as an acoustic baffle and knows how to position itself in the self-cut hole to improve the audibility of its call.

Indeed, the small tree cricket with a wing length of ~0.9 cm which produces a sound of wavelength ~11 cm faces several biophysical problems which it must resolve in order to produce a loud sound. It faces the problem of synchronising the vibration of its two wings with each other, which crickets generally solve by using a phase shifting mechanism (Montealegre-Z et al., 2009, JEB). The other is of sound cancellation cause by the anti-phase sound emanating from the front face and back face of the synchronously vibrating wings, which it solves by using a baffle that prevents this sound cancellation by avoiding this form of acoustic ‘short-circuiting’.

This remarkable finding is likely to be of broad interest to researchers studying the evolution of animal behavior, as well as behavioral ecology, neuroethology, and acoustics.Essential revisions:Abstract, second to last sentence and several other locations; I find the use of the word materials here misleading, because it implies the animals select from materials with different material properties. Instead it seems more precise to discuss how the animals choose between different possible baffle hole geometries.

This is a useful observation. We do not wish to talk about material properties such as hardness, density or weight of the leaf. The functional property that is important to the baffle is leaf size and size is what we want to refer to. We have now altered the wording, making the size aspect explicit by using the following phrase: “…they selected the best sized object and modified it appropriately to make a near optimal baffle.” The second part of the sentence, ‘modified it appropriately’ refers to baffle hole geometries.

The complete statement is explained further in the text: “Thus, given the opportunity males would indeed optimize the SRE of their baffles in terms of both selecting the appropriate materials, i.e. the largest leaf and then modifying it appropriately i.e. cutting a hole of acoustically optimal size (Figure 3).”

Here, since size is now explicitly mentioned, the terms of reference from the Abstract are now fully explained.

We also use the word material in a general description of optimization, where we believe using the term material is reasonable. We have added ‘functionally’ to make it more explicit. It now reads: “Optimization, the ability to select the functionally best material and modify it appropriately for a specific function, implies flexibility and is thought to be incompatible with inherited behaviour.”

Another passage is modified to avoid the issue raised by the referees: “Thus, by finding the best materials and making a near optimal baffle, a male tree cricket can increase his active acoustic area by as much as four times (12 dB), greatly increasing his chances of attracting a mate.” We modify this statement to now read “Thus, by finding the largest leaf and making a near optimal baffle, a male tree cricket can increase his active acoustic area by as much as four times (12 dB), greatly increasing his chances of attracting a mate.”

Introduction, second paragraph and several other locations; The advertisement sounds made by tree crickets are typically not considered to be songs; Only a very lax definition of song (e.g. in Broughton 1963; "Sound of animal origin which is not both accidental and meaningless") includes these sounds. We recommend to term them calls.

We agree and we have now changed *songs* to *calls* throughout the text of our manuscript and in the Materials and methods. Additionally, for semantic consistency, we have changed the verb form, *singing/sing* to *calling/call* throughout the manuscript.

Subsection “Singing effort and wing speed”, throughout source levels are reported at 200mm which is a less conventional reference distance. Please also report source levels with respect to 1 m distance if possible or discuss in the Materials and methods why 200mm is the better choice if you have a specific compelling reasoning.

It is true that the convention in animal bioacoustics is to give the SPL reference at 1m. When dealing with invertebrate calls, a smaller reference distance is often used, especially so for high frequency species (Mhatre and Balakrishnan, 2006; Deb, Balakrishnan, 2014). There are several reasons. One is due to the high level of attenuation calls may face in field conditions. Even under lab conditions, a smaller distance is preferred due to lower call SPLs and the very directional and ‘lobed’ pattern of sound radiation.

Tree crickets specifically are about an order of magnitude smaller (wings ~1 cm) than the wavelength of sound they produce (~11 cm at 3100 Hz). The result of this is that the sound source is directional (sometimes called ‘lobed’), and that the SPLs they produce are low (Figure 2). Even with a baffle, at 200 mm or 20 cm, the SPL from a calling male was ~60 dB SPL on average. At 1m, the call will be at ~46 dB SPL, which is close to the ambient noise which is generally ~40 dB SPL. In addition, because of a strongly lobed pattern, measuring SPL even slightly off-axis, will cause larger errors at larger distances, making the measurement un-representative of the source SPL. Such small angular deviations are difficult to control for when making measurements from a calling animal which is free to move and reposition itself. Finally, since we use two microphones, in front of and behind the singing male, the reference length is constrained by the size of the acoustic booth and the optical table.

Thus, the 20 cm distance is chosen for being physically achievable, outside the so-called near-field (> 1λ), having a good signal to noise ratio, and for being representative of the source SPL despite small angular deviation from the true axis of the sound radiator.

Subsection “Optimization of baffle acoustics”, second paragraph. Figure 3 is somewhat confusing. The leaf distribution and probability seem to belong to the left axis and SRE to the right axis; this could be made more intuitive by a different color scheme. Also, SD error bars are missing on blue baffle manufacture bars. It might be clearer if the raw data of leaf size and probability for the individuals are added.

We apologize for the complexity of this figure. Thank you for the suggestion to coordinate the colour of the axis and the line graphs, which will now make the figure clearer. We have changed the axis and the SRE lines all to red and edited the caption to reflect this.

For the baffle manufacture bars, we have the proportion of 51 individuals that made a baffle. Similarly for the leaves, we have the proportion of leaves in each size class from 570 randomly selected leaves. As such there are no SD/error bars for these data. We apologize for any confusion. To make this explicit, we have changed probability to proportion in the graph. In the legend, we have made the following changes: “(B) Blue bars depict the proportion of males that made baffles on different leaf sizes in a no-choice experiment. Grey bars depict the distribution of natural leaf sizes by depicting the proportion of leaves that fall into different size classes (N=570 leaves). The two red lines depict the SRE associated with baffled singing (solid line), and with unbaffled singing from the leaf edge (wings parallel to the leaf surface, stippled line) at different leaf sizes.”

Subsection “Progressive versus single shot optimization”, second paragraph, mole crickets optimize their burrow structure to match call frequency resonance. This contradicts the statement or gist in the Abstract that there are no precedents of optimization in innate behavior, please discuss in the manuscript with accompanying references to frame the novity of the finding in this study better.

In that passage, we do not mean to imply that there are no precedents in optimization in innate behaviour, and mole crickets truly are an excellent example of such behaviour. Yet, the inflexibility of insect behaviour remains a widely held view, for instance, see the contemporary article by Hunt, Gray and Taylor (Hunt, Gray and Taylor, 2013). To avoid confusion, within the word limit of the Abstract, we have changed the line “Optimization, the ability to select the best material and modify it appropriately for a specific function, implies flexibility and is usually thought to be incompatible with inherited behaviour.”

Within the manuscript, we discuss mole crickets extensively (Introduction, second paragraph, subsection “Progressive versus single shot optimization”, second paragraph and subsection “The mechanism for baffle optimization”, third paragraph) and the following references in our reference list are about mole cricket burrow use and optimization: Daws, Bennet-Clark and Fletcher, 1996; Bennet-Clark, 1987; Hill, Wells and Shadley, 2006; Hill, 1999.

Specifically, the entire Results section “Progressive versus single shot optimization” is inspired by a comparison of the differences between mole cricket and tree cricket optimization behaviour. Mole crickets improve their burrow resonance gradually, over about 50 minutes, whereas tree crickets optimize their baffles in a single attempt, and do not often make further modifications.

In this section, we explore the reasons behind this, consider both the properties of baffles in comparison to burrows and explore how their acoustics would be changed with further modification using a series of FE models. In the discussion, we further consider the differences between the two optimization behaviours with respect to the sensory feedback mechanism (subsection “The mechanism for baffle optimization”, third paragraph). We suggest that the reason behind the slightly suboptimal performance of mole cricket burrows in comparison to heuristic optimization may be due to Weber’s law and reference an excellent recent paper which shows how Weber’s law limits optimization in real ecological systems.

In short, tree crickets are novel in using a low-error heuristic mechanism for optimization, which allows them to achieve near optimal performance in a single attempt.

Subsection “Progressive versus single shot optimization”, second paragraph and subsection “The mechanism for baffle optimization”, last paragraph. Based on the evidence it cannot be fully excluded that males use the experience they gained in making baffles. E.g. just a few iterations could aid in increasing efficiency, although we would not know how they evaluate this. To be conclusive it would be helpful to control for the baffling history of the males by using naïve ones etc. This would be a rather simple future experiment presenting a very interesting finding. Please discuss your perspective on this in the manuscript for completeness and for stimulating further research. We are not asking for these controls as the contribution of the present work is sufficient to merit publication.

This is true. It is not possible to eliminate this possibility based on the current evidence and further experiments would be required to establish this fully. We have changed the text in this paragraph to reflect this consideration:

“It is possible that the tree crickets learn the SRE landscape, however, we believe that this is unlikely since the opportunity for learning is small due to life-history and time constraints. […] Another possibility is that they can abstract these general rules from a few trials. While we believe that inherited heuristic optimization is a more parsimonious explanation, repeating the baffle choice experiments with naïve males would provide a more definitive answer.”